# T* Revise Attenuation Tomography for Q Estimation

**Ziqi Jin, Ruoteng Wang and Ying Shi ***

School of Earth Sciences, Northeast Petroleum University, Daqing 163318, China; jinseismic@outlook.com (Z.J.);
208003010616@stu.nepu.edu.cn (R.W.)

* Correspondence: shiying@nepu.edu.cn

**Abstract:** Seismic attenuation is often calculated by attenuated travel time tomography. The accuracy of this method is controlled by the precision of attenuated travel time. In this paper, a novel T* revise in attenuated travel time tomography method for Q inversion was developed. The attenuated travel time was calculated from seismic data by using a logarithmic spectral ratio inversion strategy. In the inversion process, multiple offset traces were used for multiple attenuated travel time calculations. The proposed method produced more accurate results compared to those of the conventional approach without the requirement of choosing an optimistic frequency band. The accuracy of the proposed method was improved by avoiding the effect of overburden. Both synthetic and real data examples prove the viability and effectiveness of the proposed method.

**Keywords:** Q estimation; attenuated travel time tomography; Q compensation migration





## 1. Introduction

Seismic waves experience amplitude loss and phase distortion due to viscoelastic properties of the subsurface media (Wang et al., 2022) [1]. It is necessary to accurately estimate seismic attenuation (always expressed as $1/Q$) for compensation of energy loss and phase distortion and reservoir prediction. This is because the Q values not only relate to the attenuation which needs to be compensated by inverse Q filtering [2–4] or Q compensation migration [5–8], but also Q is related to the variation in reservoir properties, such as rock porosity and pore fluids [9–11].

Depending on the application domain, the existing methods for calculating the interlayer Q value can be divided into three categories: firstly, the time-domain Q value estimation methods; secondly, frequency domain Q value estimation methods; thirdly, the time-frequency Q value estimation methods. In the time domain, the methods are the pulse amplitude decay method, pulse rise time method, wavelet simulation method, resolved signal method, etc. Stacey and Gladwin (1974) [12] proposed the rise time principle based on the widening of the frequency band of seismic waves; the particular advantage of the rise time method is that the required record length is very short. KJartansson and Biai (1979) [13] further studied this basis and proposed a new algorithm for calculating the attenuation factor, called the rise time method. This method simplifies the implementation conditions but the error in the attenuation factor calculated by the slope is larger. A linear model for the attenuation of waves is presented in this method, and the Q value calculated is exactly independent of frequency. Jannsen (1985) [14] proposed a wavelet simulation method to obtain the optimal Q value by finding the maximum correlation between the orthogonal simulation signal and the actual detection signal. Wang et al. (2001) [15] processed wave velocity imaging and rise time imaging simultaneously through the attenuation imaging method in the time domain to efficiently calculate Q value. This method can remove the effect of source wave from the calculation process.

Tonn (1989) [16] has compared and analyzed several common Q estimation methods, including amplitude decay, analytic signal, wavelet simulation, phase simulation, spectrum simulation, rise time, pulse amplitude, spectral ratio, and spectral simulation; no one

method is generally superior. The time domain Q estimation methods are not recommended because they have some problems in distinguishing the intrinsic attenuation from other types of attenuation. It is difficult to distinguish the intrinsic attenuation from other types of attenuation. Q estimation methods developed in frequency domain are widely used as an advantage of their ability in avoiding the effects from non-intrinsic attenuation. These transform the reflections into the frequency domain, and the estimated Q values are based on the variation of their amplitude spectra. The spectral ratio method (SR) estimates Q values between large events, which is one of the most commonly used Q value estimation methods. However, when seismic data contain noise, the spectral ratio method has poor stability and the result depends on the selected frequency band. Kan (1982) [17] improved this spectral ratio method by considering the relationship between intrinsic attenuation and geometric diffusion attenuation. Raikes and White (1984) [18] established the transfer function of the attenuation factor in the frequency domain by using the fitting technique, which improved the accuracy of the calculation of the attenuation factor. Dasgupta and Clark (1998) [19] used the spectral ratio method to estimate Q values in the pre-stack reflection. The Q value was estimated in the wave channel set by establishing the relationship between the wavelet log spectrum and the Q value. The spectral ratio method has been subsequently improved by many other authors. Haase and Stewart (2003) [20] showed that the automatic selection of available frequency bands is key to the accuracy in the spectral ratio method. Hackert and Parra (2004) [21] used reflection coefficients from well logs to correct for the interlayer interference problem of reflected waves. The success of this method depends on the local relation between the well and the seismic data. Further, it needs the time windows long enough to bracket several periods of the peak frequency of the seismic waves. Gurevich and Pevzner (2015) [22] analyzed the error in Q value estimation by the spectral ratio method under the assumption that the Q value varies with frequency. Cao et al. (2014) [23] proposed the logarithmic spectral root equation method for Q inversion, which has improved accuracy and noise immunity. Wang et al. (2015) [24] proposed the logarithmic spectral area difference method. When seismic wave propagates through the media with a decrease in amplitude and reduction in bandwidth, it leads to the variation of the logarithmic spectral area of seismic wave, which enhances the stability of the algorithm compared to that of the traditional spectral ratio method. Guo et al. (2018) [25] proposed a Q value estimation method combining Capon2D and weighting strategy to improve noise immunity and suppress wavelet and relate Q value estimation results to reservoir fracture development characteristics to guide shale gas exploration.

Compared with the conventional spectral ratio method, the improved method is insensitive to high-frequency noise and has the ability to suppress wavelet interference. Stable Q value estimation results were obtained by Sangwan et al. (2019) [26] through the nonlinear inverse amplitude spectral ratio method. To address the problem that the pre-stack seismic data have complex ray paths and there are noise and tuned interference effects that make Q calculation difficult, Liu and Li (2020) [27] used the local information of seismic events combined with spectral ratio method and multi-ray waveform spectrum to jointly invert Q values based on the seismic wave ray propagation principle, and they also verified the validity of the calculation results by predictive mapping technique. Based on the spectral ratio method, Jin Zhang and Guoshu Zhang (2022) [28] performed the second-order Taylor series expansion of the seismic wave amplitude decay term, established the equation related to Q value by the difference of amplitude spectrum at different moments, and proposed a Taylor series expansion-based amplitude spectrum integral difference method, which effectively improved the noise immunity of Q value estimation. In response to the disadvantages of poor stability and dependence on frequency band selection of the spectral ratio method, Yang and Liu (2022) [29] proposed the weighted spectral ratio method with reference to the idea of the weighted center-of-mass frequency shift method, which effectively enhanced the stability of Q value prediction. Quan and Harris (1997) [30] used the shift of centroid frequency to estimate Q values for VSP data, which is called

the centroid shift method. Yan (2001) [31] used the centroid frequency shift method to combine the interwell seismic velocity and Q value joint tomography on this basis. Li and Wang et al. (2016) [32] combined the traditional center of mass frequency shift method and Gaussian weighting coefficients to propose a weighted center of mass frequency shift method, which reduces the influence of band selection and improves the accuracy and robustness of Q value calculation. Based on a similar idea, Zhang and Ulrych (2002) [33] brought up the peak frequency method for Q estimation and assumed the source wavelet is Ricker-like. Wang and Gao (2018) [34] introduced the generalized seismic wavelet function on top of the peak frequency shift method to establish the relationship between the GSW function and the quality factor, which enhances the noise immunity of Q value prediction. Q estimation methods based on time frequency domain mainly use wavelet transform, S transform, and the generalized S transform and other time frequency analysis methods, including wavelet envelope peak frequency offset method. Constant Q can be estimated via Gabor Analysis (CVG) based on Gabor spectrum, etc. Reine et al. (2009) [35] compared four time-frequency analysis methods, namely the short time Fourier transform (STFT), the Gabor transform, the S transform, and the continuous wavelet transform, and pointed out that different transforms have different time frequency characterization capabilities. A time frequency characterization method has also been proposed using least square inversion and regularization constraint algorithms for seismic data sand detection, noise suppression, and Q value estimation of post stack seismic data. Based on the wavelet envelope peak frequency migration method, Zhao and Gao (2013) [36] proposed the EPIFVO method to calculate the quality factor of pre-stack CMP data by using horizon information, which provides an effective basis for reservoir prediction. Based on the generalized S transform, Wu and Xu et al. (2018) [37] proposed the continuous spectral ratio slope method to calculate the Q value of the pre-stack CMP channel set based on the linear relationship between Q value and offset distance and the Dix formula, which improves the resolution of the Q profile and correctly characterizes the absorption characteristics of seismic wave energy by the formation. Gao and Wei (2020) [38] proposed a Q estimation method based on the mutual correlation function and S transform to calculate the Q value based on the linear relationship between the spectral ratio and frequency, which can reduce the error caused by the Gaussian window function. Xu and Gao (2022) [39] proposed a method to extract the quality factor Q based on the S transform and variational method to address the problems of insufficient noise immunity and over dependence on the seismic wavelet type of the traditional method. Each of these methods has its own advantages and disadvantages, and none of them is universally applicable. The effectiveness depends on the quality of the records.

In earthquake seismology, the attenuated travel time tomography is first applied for attenuation calculation in the shallow crustal [40]. In seismic exploration, the approach is modified and used for seismic data attenuation calculation [41–43]. The Q volume can be calculated by this method and its resolution is controlled by the size of the grid. By comparing the estimated and real attenuated travel time, the Q volume can be inverted [44]. The key of the method is the accuracy of attenuation travel time. The conventional method calculates it by choosing frequency components with high S/N ratio and always introduce bias of the results. To overcome this disadvantage, the proposed method uses spectral ratio of seismic data in simultaneous inversion. The method can calculate multiple attenuated travel times simultaneously. The error from frequency component selection can be avoided by taking into account the different ray paths in the overburden and thus give more accurate Q results.

## 2. T* Revise Attenuated Travel Time Inversion

Q value can be calculated by using the inverted attenuated travel time [45]. The misfit of attenuated travel time between the travel time from the recorded data and synthetic data can be used for Q value calculation; Q value is updated by minimizing the misfit:

$$\delta t_i^* = t_r^* - t_m^* = \sum t_{ij} \delta Q_j^{-1},$$ (1)

where $t_r^*$ and $t_m^*$ are the attenuated travel times calculated from real data and synthetic data, respectively. $\delta t_i^*$ is the difference between the attenuated travel time from real data and $\delta Q_j^{-1}$ the attenuated travel time from synthetic data, and is the Q value needed to be updated towards the true value.

The attenuated travel time calculated from real data $t_r^*$ can be calculated by the logarithmic spectral ratio (LSR) inversion method:

$$b = ln\left(\frac{A_{(t_2,f)}}{A_{(t_1,f)}}\right) = ln(G) - \frac{\pi f(t_2 - t_1)}{Q} = B - \pi f t_r^*,$$ (2)

where A is amplitude spectra of seismic wave; at travel times $t_1$ and $t_2$, the amplitude spectra of seismic waves are $A_{(t_1,f)}$ and $A_{(t_2,f)}$, respectively. The frequency independent attenuation is G. B is the intercept term and equals to *ln(G)*. The proposed method can inverse both *ln(G)* and $t_r^*$ using multi traces to avoid the effect of frequency range selection in the conventional LSR method. When the seismic events are close to each other in the vertical direction, the time–frequency spectrum can be obtained by time–frequency transformation of the seismic signal. The common time–frequency analysis methods include short-time Fourier transform, continuous wavelet transform, Wigner–Ville distribution, S-transform, generalized S-transform, etc. Short-time Fourier transform is limited by the Heisenberg inaccuracy and the window function, and the time–frequency resolution is fixed and cannot be improved simultaneously. Continuous wavelet transform needs to transform time–scale distribution into time–frequency distribution, which can adjust the window function adaptively with frequency and solve the shortcoming of fixed short-time Fourier time–frequency resolution. The Wigner–Ville distribution portrays the time–frequency distribution of the signal from the perspective of the energy spectrum or power spectrum and does not use the window function, thus avoiding the interaction between the time and frequency domains and has a high time–frequency resolution, but when dealing with multi-component signals, the interference of cross terms is generated. The S-transform uses a Gaussian window function to directly convert the signal from the time domain to the time–frequency domain, with variable time–frequency resolution. The window function of S-transform varies with frequency in a fixed trend, which lacks flexibility. The generalized S-transform is able to adjust the window function by introducing the adjustment factor, which can adjust the time–frequency resolution according to the signal characteristics and has good practicality, as follows:

$$GST(\tau, f) = \int_{-\infty}^{+\infty} f(t) \frac{\lambda |f|^p}{\sqrt{2\pi}} e^{\left(-\frac{\lambda^2 f^{2p}(t-\tau)^2}{2}\right)} e^{(-i2\pi f t)} dt$$ (3)

where $\lambda$ and $p$ are the scale adjustment factors of the Gaussian window function.

The different ray travel paths from the selected reflections will cause error in $t_r^*$ results. In each layer, the different travel times of the selected reflections is taken into account. Thus, the overburden effect can be effectively avoided by the proposed method. As shown in Figure 1, a two-layer model is used here to test the performance of the proposed method. By using the reference reflections $A_1, A_3, \ldots, A_{2n-1}$ from the first layer and target reflections $A_2, A_4, \ldots A_{2n}$ from the second layer, the proposed method calculates attenuated travel time simultaneously.

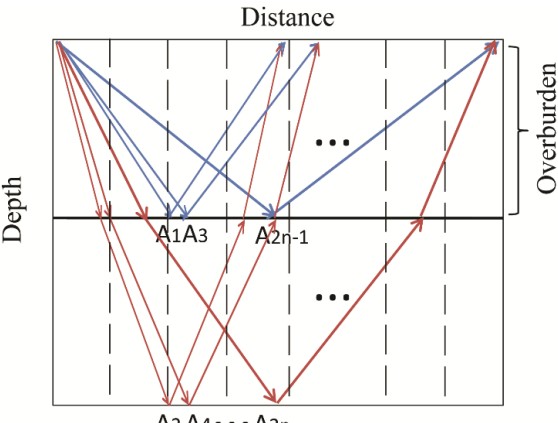

**Figure 1.** The reference and target reflections in a two-layer model. In this model, the overburden is the first layer, which is the layer above the target layer, that is the second layer.

The logarithm ratio of amplitude spectra of reflections $A_1$ in the overburden and $A_2$ in the second layer are used for attenuated travel time $(t^*_2 - ^*_1)$ calculation; also, reflections $A_1$ and $A_3$ in the overburden are used for attenuated travel time $(t^*_3 - t^*_1)$ calculation. Generally, reflection $A_{2n-1}$ in the overburden and $A_{2n}$ in the second layer are used for attenuated travel time $(t^*_{2n} - t^*_{2n-1})$ calculation; also, reflections $A_{2n+1}$, $A_{2n-1}$ in the overburden are used for attenuated travel time $(t^*_{2n+1} - t^*_{2n-1})$ calculation:

$$\begin{cases} b_1 = ln(\frac{A_2}{A_1}) = B_1 - \pi f\left(t^*_2 - t^*_1\right), \\ b_2 = ln(\frac{A_3}{A_1}) = B_2 - \pi f\left(t^*_3 - t^*_1\right), \\ b_3 = ln(\frac{A_4}{A_3}) = B_3 - \pi f\left(t^*_4 - t^*_3\right), \\ b_4 = ln(\frac{A_5}{A_3}) = B_4 - \pi f\left(t^*_5 - t^*_3\right), \\ \qquad\qquad \vdots \\ b_{2n-1} = ln(\frac{A_{2n}}{A_{2n-1}}) = B_{2n-1} - \pi f\left(t^*_{2n} - t^*_{2n-1}\right), \\ b_{2n} = ln(\frac{A_{2n+1}}{A_{2n-1}}) = B_{2n} - \pi f\left(t^*_{2n+1} - t^*_{2n-1}\right), \end{cases} \tag{4}$$

where $b_1$, $b_2$, $b_{2n}$ are spectra ratios calculated from the reflection pair of $A_1$ and $A_2$, pair $A_1$ and $A_3$, pair $A_{2n+1}$ and $A_{2n-1}$, respectively. Then we modify and re-write Equation (4) as:

$$\begin{pmatrix} b_1 \\ b_2 \\ \vdots \\ b_{2n} \end{pmatrix} = \begin{pmatrix} v & z & \cdots & z & z & -\pi F & -\pi F & z & \cdots & z & z & z \\ z & v & \cdots & z & z & -\pi F & z & -\pi F & \cdots & z & z & z \\ \vdots & \vdots & \ddots & \vdots & \vdots & \vdots & \vdots & \vdots & \ddots & \vdots & \vdots & \vdots \\ z & z & \cdots & v & z & z & z & z & \cdots & -\pi F & -\pi F & z \\ z & z & \cdots & z & v & z & z & z & \cdots & -\pi F & z & -\pi F \end{pmatrix} \begin{pmatrix} B_1 \\ B_2 \\ \vdots \\ B_{2n} \\ t^*_1 \\ t^*_2 \\ \vdots \\ t^*_{2n} \end{pmatrix}, \tag{5}$$

where vectors $v = (1\ 1 \ldots 1)^T$, $F = (f_1\ f_2 \ldots f_n)^T$, and $z = (0\ 0 \ldots 0)^T$. By solving Equation (5), the terms B and $Q^{-1}$ are inverted. To update $\delta Q^{-1}$, the inverted attenuated travel times are then used in Equation (1):

$$\delta t_{12}{}^* = (t^*_{r1} - t^*_{r2}) - (t^*_{m1} - t^*_{m2}) = \sum(t_1 - t_2) \cdot \delta Q^{-1}, \tag{6}$$

where $t^*_{r1}$ and $t^*_{r2}$ are the calculated attenuated travel times from real data, and $t^*_{m1}$ and $t^*_{m2}$ are calculated attenuated travel times from synthetic data. The final Q can be derived when the error in $\delta t_{12}{}^*$ is acceptable.

### 3. Synthetic Data Test

This paper assumes that the velocity is known, and the proposed method is applied on a layered model to test its viability and efficiency. Velocities, densities, and Q values in each layer are listed in Table 1. A shot gather is generated by ray tracing with a Ricker wavelet of 60 Hz peak frequency. The velocity can be estimated by tomographic inversion or refraction method in field data. Tomographic inversion is divided into two categories: diving wave tomography and waveform tomography [46]. Theoretically, the waveform tomography inversion algorithm has higher velocity inversion accuracy, but the method is more dependent on the initial model and seismic low-frequency data, so its application is limited. Research on diving wave tomography inversion has shifted from mid-deep models to near-surface models in recent years, gradually weakening the dependence on stratigraphy and moving toward a data-driven direction. Refractive wave method includes the wave front method, delay time method, intercept time method, t0 difference method, and generalized reciprocity method. The principle of the method is to calculate the layer thickness and velocity of each layer from the travel time data of direct and refracted waves [47]. The constant Q model is used for velocity dispersion and attenuation calculation of seismic wave (Figure 2).

**Table 1.** Parameters for layered model.

| | Depth /m | Density /(kg·m$^{-3}$) | Vs /(m·s$^{-1}$) | Vp /(m·s$^{-1}$) | True $Q$ | Inverted $Q$ |
|---|---|---|---|---|---|---|
| 1 | 200 | 2095.2 | 1200 | 2100 | 50 | 49.991 457 |
| 2 | 200–400 | 2143.3 | 1300 | 2500 | 100 | 100.001 44 |
| 3 | 400–600 | 2188.5 | 1400 | 2700 | 200 | 199.972 46 |
| 4 | 600–800 | 2290.6 | 1500 | 3000 | 300 | |

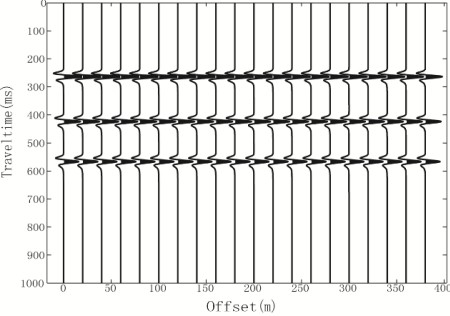

**Figure 2.** Synthetic shot gather of the layered model. The move out of the traces are corrected.

Figure 3 shows the calculated attenuated travel time by using the proposed method and the conventional attenuated travel time tomography method. The proposed method produced more accurate attenuated travel time results. With the increase of the offset distance, the conventional method will gradually increase the attenuated travel time, and compared with the proposed method, the error in the attenuated travel time is large. For the proposed method, we can see that when the offset distance is about 20, the error between it and the true value is the smallest, and as the offset distance increases, the error gradually becomes larger, and when the offset distance reaches the maximum, the error between it and the true values also reaches the maximum. The error between the proposed method and the true value is within an acceptable range, so this method is considered to be reliable. Then the Q value is estimated by using Equation (4), and the initial Q value is set to 100. The Q value results are shown in Figure 4. The conventional attenuated travel time tomography method contains error in attenuated travel time and results in a bias of Q value results, especially for the shallow layer. In the first layer, the error in the Q value results is small by the conventional attenuated travel time tomography method. Equivalent Q values are calculated in the first layer by horizontally selecting the seismic reflection waveform, so

there is no additional influence from the overburden. The overburden effect comes from the attenuation effect of different travel paths in the overburden of the reflections from A1 and from A2 seen in Figure 1. This cumulative attenuation effect cannot be eliminated and leads to errors in Q estimation in the conventional LSR method. While in the deeper layers, the Q value is calculated by longitudinally selecting the seismic reflection waveform. Especially the second layer contains more effects from the overburden and thus leads to error in the conventional attenuated travel time tomography method. The proposed method gives Q value results close to the true values.

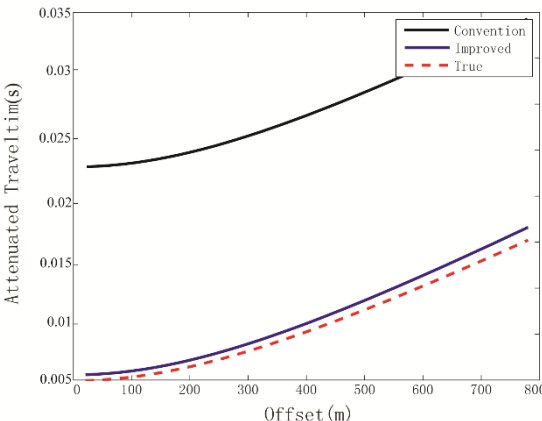

**Figure 3.** Comparison of attenuated travel times by using the proposed method and conventional attenuated travel time tomography method.

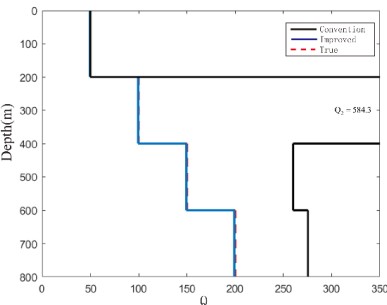

**Figure 4.** Comparison of Q value estimation results calculated using the proposed method and conventional attenuated travel time tomography method.

Then the Q value in the first layer are made to vary in the horizontal direction. The Q value estimation results in the second layer of the proposed method and the conventional attenuated travel time tomography method are compared (Figure 5). Due to the error in attenuated travel time, the Q value results of conventional attenuated travel time tomography method contains large errors, while the proposed method gives more accurate Q value results. When the offset distance is less than 300 m, the error between the conventional method and the true value is about 15%, and when the offset distance is more than 300 m, the error in the conventional method increases dramatically, and when the offset distance is 800 m, the error in the conventional method reaches 60%. In contrast, the improved method proposed in this paper is almost independent of the offset distance, and the Q value estimation results are always approximate to the true value.

Random noise is then added to the model data for noise immunity test. The Gaussian random noises are 5%, 10%, and 15% of maximum amplitude, and the Q value results are shown in Figures 6–8. The proposed method gives more accurate Q results compared with that of the conventional method, especially for 15% Gaussian random noise. Both the conventional method and the proposed method in this paper have small errors in the first layer. This is because we selected the seismic reflection waveform horizontally to avoid the

effects of the overburden. In the second layer, the influence of the overburden is received more, and the Q value calculation result of the conventional method far exceeds the true value, and the calculation error in the proposed method in this paper is still small. In the third and fourth layers, with the increase of Gaussian noise, the error in the conventional method increases exponentially, while the calculation error in the proposed method is still within the acceptable range with strong robustness and high accuracy.

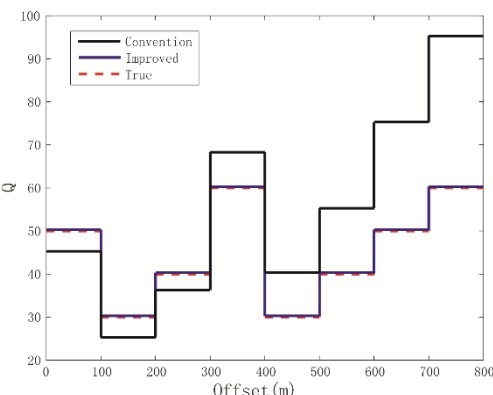

**Figure 5.** Comparison of Q value results estimated in the second layer calculated using the proposed method and conventional attenuated travel time tomography method.

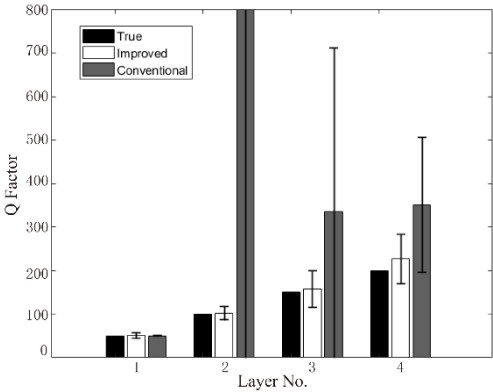

**Figure 6.** Comparison of average Q with 5% Gaussian random noise calculated using the two methods with true Q values. Standard deviation of Q are also plotted.

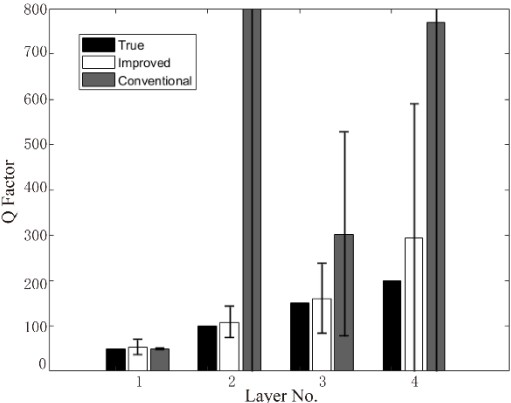

**Figure 7.** Comparison of average Q with 10% Gaussian random noise calculated using the two methods with true Q values. Standard deviation of Q are also plotted.

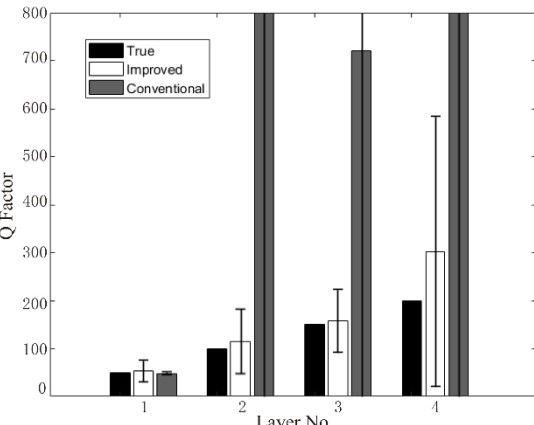

**Figure 8.** Comparison of average Q with 15% Gaussian random noise calculated using the two methods with true Q values. Standard deviation of Q are also plotted.

## 4. Field Data Application

The proposed method is then applied to field data from the northern Chinese oil field. The Q of the layer between travel times of 3.2 s and 4.0 s (Figure 9) is calculated. The subsurface is grided by 80 × 80 and each grid is 80 m.

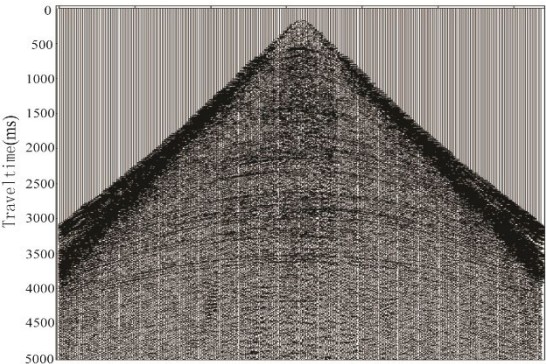

**Figure 9.** Field data of common shot gather.

The inversion Q results are shown in Figure 10. The Q inversion results by the proposed method (Figure 10b) shows a higher quality compared with that of the conventional method, such as the artifact at depth 4000 m and also the background noise in from depth 7000 m to 8000 m.

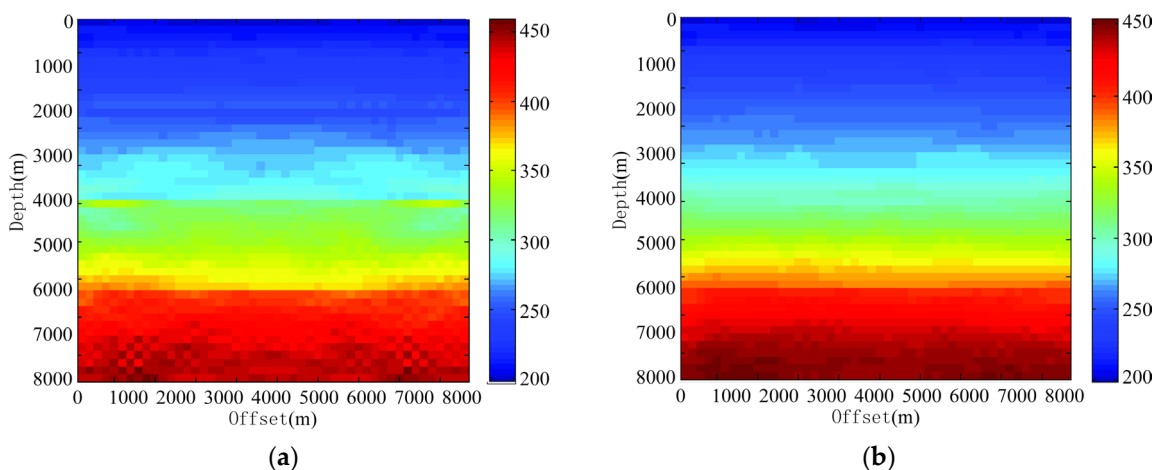

**Figure 10.** *Q* profiles calculated by using conventional method (**a**) and the proposed method (**b**).

Then the data are processed by Q compensation migration with estimated Q as the input. The common reflection point gatherings from the two methods are compared in Figure 11. The results of the proposed method give better recovered amplitudes and corrected distortions of phase. The attenuated reflections below 3.5 s are recovered. Significantly higher horizontal continuity and vertical resolution between 3.44 s and 3.5 s can be found in Figure 11 compared to conventional methods. The resolution of reflections around 3.6 s are higher when compared with that of the conventional method.

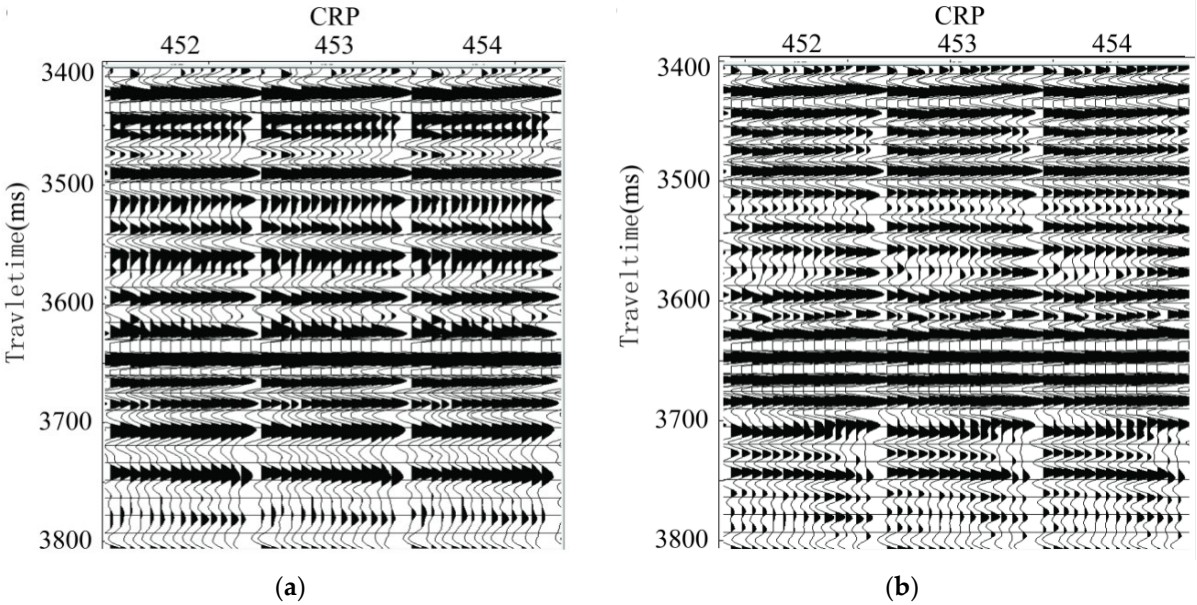

**Figure 11.** Comparison of Q compensation migration gathers by using the Q results of (**a**) the conventional method and (**b**) the proposed method.

## 5. Conclusions

A novel T* revise attenuated travel time inversion method is introduced for Q inversion. The accuracy of attenuated travel time, which is key in attenuated travel time tomography, is improved by the proposed method. Thus, more accurate Q results can be inverted with more details of the longitudinal and transverse variations of Q profiles. The proposed method improves the stability of the inversion results by using multiple reflected wave information to simultaneously invert the multiple attenuation travel times. The logarithm ratio of amplitude spectra of reflections in the overburden and in the second layer are used for attenuated travel time difference between the two layers calculation; also the two reflections in the overburden are used for attenuated travel time difference in the overburden calculation. In addition, the proposed method uses frequency components with good S/N for calculation and avoids the effect of choice of an optimistic frequency range in conventional method, which will introduce error of *Q*.

The overburden effect comes from the attenuation effect of different travel paths in the overburden. This cumulative attenuation effect cannot be eliminated and leads to error in Q estimation in the conventional LSR method. In the proposed method, the overburden effect is avoided by accounting for the travel time differences in reflections in each layer. The effectiveness is proved by synthetic data and field data application. Q compensation migration can give more reliable data using the Q results from the proposed method. The proposed method assumes that the velocity is known before Q inversion. For further studies, velocity and Q inversion can be considered and updated simultaneously.

**Author Contributions:** Conceptualization and methodology, Z.J.; writing—original draft preparation, Z.J. and R.W.; writing—review and editing, Z.J. and R.W.; supervision and funding acquisition, Y.S. All authors have read and agreed to the published version of the manuscript.

**Funding:** This research was funded by the National Natural Science Foundation of China, grant number 42274173.

**Institutional Review Board Statement:** Not applicable.

**Informed Consent Statement:** Not applicable.

**Data Availability Statement:** Not applicable.

**Acknowledgments:** The authors are grateful to the National Natural Science Foundation of China (grant no. 42274173) for supporting this research. Thanks also are due to Wei Xu for his valuable suggestions and discussions.

**Conflicts of Interest:** The authors declare no conflict of interest.

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
