# Peer review of "T* Revise Attenuation Tomography for Q Estimation"

_applsci, doi:10.3390/app13085201_

Round 1

Reviewer 1 Report

This paper develops a new T* revise in the attenuated travel time tomography method for Q inversion. The proposed approach utilizes a logarithmic spectral ratio inversion strategy to calculate the attenuated travel time from seismic data, using multiple offset traces in the inversion process. The method proposed by the authors yields more accurate results than the conventional ones without requiring selecting an optimistic frequency band. Additionally, their work improves accuracy by avoiding the overburden effect.

The work is well-written and presents an extensive bibliographic review of existing methods and their applications.

The paper presents clear motivations, methodologies, and results. Figures 3, 4, and 5 show comparisons between results from the new and conventional models, showing a considerable improvement when using the new proposed model.

I have not found any significant flaws, and I think the present work has enough material to be published in Applied Sciences. In this way, I recommend publishing the present paper after some minor revisions described below.

  • Section 4 should be improved by discussing the real data results better. The results are presented, but the discussion needs to be more specific. 

  • The same thing happens in the “Conclusions” section. The conclusions regarding the results from actual data could be elaborated more.

Reviewer 2 Report

Dera Authors

Scientific comments

It is an interesting work about seismic attenuation, but the reviewer has reservations; you should pay attention to item 1. It must describe Q and integrate the seismic concepts of reflection and refraction; e.g. recording of shear (S) and compression (P) waves require a source of seismic energy that generates seismic pulses;

For a better interpretation of Figure 1, Table 1 and the model, you should think about; (e.g.) The waves received by the geophones are those that travel directly in the upper layer or waves that travel on top of the first layer with a critical angle of incidence (ic) with velocity V1, refract at the interface with the velocity of that medium (V2) and appear at the surface with velocity V1. In the geophones closest to the source, the first waves to arrive are the direct ones and define a branch in the dromochronic that passes through the origin and the inclination is the inverse of the propagation velocity (1/V1) at the upper interface. As the reception distance of the arrival time increases, the critical distance (Xc) is reached, corresponding to the arrival of the first refracted waves, which 1/V2 means the propagation velocity in the lower stratum. The corresponding dromochronic does not pass through the origin and its intersection with the time axis means the intercept time (Ti). The definition of Xc and Ti makes it possible to establish formulas for calculating the velocity of propagation of P waves in the two different layers and the depths of the upper medium. So successively…

The measurement of the velocity of propagation of S waves were made with which intervals? What are the interpretive methods for evaluating the propagation velocity of shear waves (Vs)? It should be noted that the signals obtained in the seismic tests close to the surface have lower quality, fundamentally motivated by the less spaced arrival in time of the signals propagated by the terrain. These are questions that can influence attenuation and should be part of the study.

Item 2 2. T* revise attenuated travel time inversion, e.g., “The logarithm ratio of amplitude spectra of reflections in both layers are used for 175 attenuated travel time calculation: You should better develop the methodology and equations used and pay attention to the description of all index variables and coefficients used in the equations. Should consider improving the understanding of the model and its application.

e.g. The improved 80 Capon2D Q… in addition to the citation, you must show your understanding and explanation. e.g. (line144)” with high S/N ratio and...” explain.

Table 1 is too far from the reference and should describe it in the integration of seismic knowledge. Density has no units. Or else you should write volume (bulk) mass - g/cm3.

The reference to figure 3 must be before this one.

Line 205 to 208, 226 and 244 …; An explanation is needed for the statement...

The bibliography is correct.

Editorial comments

Improve some figures;

Line 90 ...[28].Based … add space

Round 2

Reviewer 2 Report

Dear Authors

In general, it satisfied the reviewer's requirements...